# The Human Mycobiome in Chronic Respiratory Diseases: Current Situation and Future Perspectives

**DOI:** 10.3390/microorganisms10040810

**Published:** 2022-04-13

**Authors:** Juan de Dios Caballero, Rafael Cantón, Manuel Ponce-Alonso, Marta María García-Clemente, Elia Gómez G. de la Pedrosa, José Luis López-Campos, Luis Máiz, Rosa del Campo, Miguel Ángel Martínez-García

**Affiliations:** 1Department of Microbiology, Ramón y Cajal University Hospital, Ramón y Cajal Health Research Institute, 28034 Madrid, Spain; juandios.caballero@salud.madrid.org (J.d.D.C.); pedromanuel.ponce@salud.madrid.org (M.P.-A.); mariaelia.gomez@salud.madrid.org (E.G.G.d.l.P.); rosacampo@yahoo.com (R.d.C.); 2CIBER of Infectious Diseases (CIBERINFEC), Institute of Health Carlos III, 28029 Madrid, Spain; 3Department of Pneumology, Central Asturias University Hospital, 33011 Oviedo, Spain; mgclemen@gmail.com; 4Principality Asturias Health Research Institute (ISPA), 33011 Oviedo, Spain; 5Medical-Surgical Unit for Respiratory Diseases (CIBERES), Institute of Biomedicine of Seville (IBiS), Virgen del Rocío University Hospital, University of Seville, 41013 Sevilla, Spain; lcampos@separ.es; 6CIBER of Respiratory Diseases (CIBERES), Institute of Health Carlos III, 28029 Madrid, Spain; luis.maiz@salud.madrid.org (L.M.); mianmartinezgarcia@gmail.com (M.Á.M.-G.); 7Department of Pneumology, Ramón y Cajal University Hospital, 28034 Madrid, Spain; 8Department of Pneumology, La Fe University and Polytechnic Hospital, 46026 Valencia, Spain

**Keywords:** mycobiome, microbiome, next-generation sequencing, fungal pathogenesis, cross-kingdom interactions

## Abstract

Microbes play an important role in the pathogenesis of chronic lung diseases, such as chronic obstructive pulmonary disease, cystic fibrosis, non-cystic fibrosis bronchiectasis, and asthma. While the role of bacterial pathogens has been extensively studied, the contribution of fungal species to the pathogenesis of chronic lung diseases is much less understood. The recent introduction of next-generation sequencing techniques has revealed the existence of complex microbial lung communities in healthy individuals and patients with chronic respiratory disorders, with fungi being an important part of these communities’ structure (mycobiome). There is growing evidence that the components of the lung mycobiome influence the clinical course of chronic respiratory diseases, not only by direct pathogenesis but also by interacting with bacterial species and with the host’s physiology. In this article, we review the current knowledge on the role of fungi in chronic respiratory diseases, which was obtained by conventional culture and next-generation sequencing, highlighting the limitations of both techniques and exploring future research areas.

## 1. Introduction

Chronic respiratory diseases (CRDs) such as asthma, cystic fibrosis (CF), non-CF bronchiectasis, and chronic obstructive pulmonary disease (COPD) are a group of diseases characterised by abnormal conditions of the respiratory system. CRDs are considered the fourth leading cause of morbidity and mortality worldwide, affecting approximately 1 billion people globally, causing an estimated 7.5 million deaths per year, and creating a massive global economic, healthcare, and social burden [1]. Conditions that occur in these diseases, such as impaired defence mechanisms, the use of immunosuppressants, and the frequent use of antibiotics, likely predispose individuals with CRD to fungal colonisation and overgrowth in their lower airways. Asthma is characterised by repeated episodic symptoms, such as shortness of breath and wheezing upon exposure to an allergen [2]. COPD is a chronic inflammatory disease in which there is an irreversible airflow obstruction that causes shortness of breath, coughing, mucus production, and wheezing [3]. Bronchiectasis is a heterogeneous and complex disorder characterised by a chronic airway inflammatory disease associated with multiple potential aetiologies, both pulmonary and systemic, which in turn lead to the destruction and irreversible dilation of the airways and to acute and chronic infection by potentially pathogenic microorganisms, including fungi [4,5,6,7,8,9]. Bronchiectasis is closely linked to COPD and asthma [10,11]. CF is an autosomal recessive disease caused by mutations in the CF transmembrane conductance regulator (CFTR) gene, which encodes a chloride and bicarbonate transporter that is mostly expressed in exocrine epithelia. The loss of CFTR function causes an electrolyte imbalance, resulting in the production of a thickened, dehydrated exocrine secretion that, at the respiratory level, impairs mucociliary clearance and allows polymicrobial colonisation of the lower airways. This chronic colonisation triggers an inflammatory response that is responsible for tissue destruction and a progressive loss of lung function. *Pseudomonas aeruginosa* is recognised as the microorganism with the greatest impact on lung function; however, the role of fungi in CF lung disease has always been controversial [12,13].

The recent use of culture-independent microbiological techniques based on deep sequencing, also known as next-generation sequencing (NGS), has shown that the respiratory tract of healthy individuals is not sterile, as formerly thought, but composed of a complex microbial community, the microbiome. Most studies on this subject have focused on the bacterial component of this microbiome, whereas other organisms such as viruses (virome) and fungi (mycobiome) have been less-investigated. The term mycobiota refers to the fungal component of a given microbial community, whereas mycobiome refers to their corresponding genomes [14]. It has been shown that the microbial communities present in the lungs of patients with CRD significantly differ from those of healthy individuals due to the disruption of microbial homeostasis, which is referred to as dysbiosis. These changes include not only those related to the microbiome composition but also changes in total microbial content as well as their abundance [14].

There is growing evidence that the lung mycobiome has a significant impact on the clinical outcome of CRD. Thanks to culture-independent methods, especially NGS, several fungi that were previously undetected by classical culture methods have been identified in human lungs. Molecular studies have shown that the structure and diversity of the lung mycobiota vary between differing populations (healthy individuals and those with various diseases) and this variation could play a role in CRD. Moreover, the interaction between the mycobiome and bacteriome and/or virome appears to be a cofactor of inflammation and host immune response, thereby contributing to the decline in lung function and disease progression [15]. In this review, we will focus on the prevalence of fungal isolation and its clinical significance in CRD and summarise the conclusions drawn from the NGS study of the lung mycobiota.

## 2. Prevalence and Clinical Significance of Fungi in Chronic Respiratory Disease

### 2.1. Chronic Obstructive Pulmonary Disease

The prevalence of fungal infection in COPD has not been as extensively studied as bacterial infection. This detection depends on whether acute or stable patients are evaluated and, above all, on the techniques used to recognise the fungi, which include fungal cultures, nucleic acid detection, sensitisation, and specific markers for specific fungal species, such as galactomannan antigen for *Aspergillus* [16,17]. The prevalence of chronic fungal infection is, therefore, variable and seldom studied. Studies have placed the prevalence of fungal infection at varying rates that are close to 20% [16,17,18,19], which represents a substantial prevalence. However, Bafadhel et al. (2011) showed that approximately 50% of stable patients with COPD at baseline had culturable filamentous fungi, 75% of which were *Aspergillus fumigatus* [20]. Of the hospitalised patients with COPD, 1.3–3.9% develop invasive aspergillosis, based on positive cultures of *Aspergillus* spp. and radiological findings [17].

Fungal sensitisation appears to play an important role in the clinical presentation and progression of COPD [21]. Again, the way in which fungal infection is studied should be considered when interpreting these results. Probably the most consistently described clinical effect is the relationship between fungal infection and the risk of exacerbations independent of COPD severity and stage [22]. A number of authors have suggested that the frequency of *Aspergillus* detection in patients with COPD might be associated with the early Global Initiative for Chronic Obstructive Lung Disease stages [16]; however, this finding requires further scrutiny. An interesting association between fungi and COPD relates to the use of inhaled corticosteroids. Various studies have shown that corticosteroid therapy is associated with increased filamentous fungal burden in allergic fungal disease [20,23]. There is increasing evidence of the importance of fungi in driving type-2-mediated immunopathology, suggesting a role for inhaled corticosteroids in treatment. The interaction between *A. fumigatus* and *P. aeruginosa* increases the fungal burden and susceptibility to bacterial infection, adding complexity to this pathological mechanism.

### 2.2. Cystic Fibrosis

In addition to chronic bacterial lower respiratory tract infection, patients with CF are predisposed to fungal colonisation, possibly due to aggressive antibiotic therapy and the repeated exposure to pathogenic organisms [24]. The prevalence of fungi in the respiratory secretions of patients with CF varies among published studies, which could be explained by the heterogeneity in study design, the lack of standardisation of mycobiological analyses, and the different geographical and environmental factors where the investigations were conducted [25,26]. *Aspergillus* spp. and *Candida albicans* are the fungi most frequently found in the airways of patients with CF, and their prevalence has increased in recent decades [27]. *A. fumigatus* is the predominant filamentous fungal species, with a prevalence ranging from 9 to 57% in respiratory isolates from these patients, followed at some distance by *Scedosporium apiospermum* and *Exophiala dermatitidis*, whose prevalences range from 8–14% and 1–16% [13,25,27,28,29]. Other less common fungi detected by culture are the genera *Lomentospora* and *Trichosporon*. *Lomentospora prolificans* (formerly *Scedosporium prolificans*) has a more restricted distribution than *S. apiospermum* and is associated with warm climates (Australia, Southern Europe, and USA), with a prevalence ranging from 0 to 3.8% [30,31]. *Trichosporon* is a basidiomycetous yeast whose prevalence is even lower, generally less than 1% [30]. Despite the low prevalence of the latter two genera, these fungi are of clinical importance in invasive disease because they are highly resistant to most antifungals [32].

The prevalence of *Aspergillus* is thought to increase with age, disease severity, and chronic antibiotic use [33,34,35,36,37]. *Aspergillus* species can produce several types of disease, such as aspergillus bronchitis, chronic necrotising aspergillosis, invasive aspergillosis, aspergilloma, asthmatic reactions (bronchial asthma, extrinsic allergic alveolitis, and allergic bronchopulmonary aspergillosis (ABPA)), as well as underlying subclinical inflammation and direct damage to the lung epithelium by fungal proteins and enzymes [33,38,39]. Given that fungi typically coexist with potentially pathogenic bacterial species in these patients’ airways, it is difficult to differentiate their pathogenic role in CF. A number of authors have postulated that *A. fumigatus*, either due to chronic infection or sensitisation to this fungus, is an independent risk factor that promotes hospitalisation, lung function deterioration, and progression of structural damage [40,41,42]. Most authors, however, agree that *Aspergillus* behaves as a colonising microorganism that does not require treatment with antifungals [43,44]. This consideration would be valid for all other filamentous fungi regularly isolated in patients with CF.

There are few studies on *C. albicans* and other yeasts in CF. Although a number of authors claim otherwise, most research on the subject shows that yeasts, particularly *C. albicans*, do not contribute to lung disease in patients with CF [27,45,46,47,48]. However, the basidiomycetous yeast *Trichosporon* might be an exception, given that it has been associated with a poorer course of lung disease and a higher frequency of exacerbations, can cause invasive disease in post-transplantation patients, and is associated with high mortality [49,50,51]. Research on the microbiome is likely to provide further insights into the risk factors that favour the acquisition of fungi in these patients’ airways and the pathogenic role of these fungi.

### 2.3. Non-Cystic Fibrosis Bronchiectasis

The prevalence of fungal colonisation in the airways of patients with non-CF bronchiectasis varies according to geographical area, microbiological culturing methodology, and aetiology of the bronchiectasis [52,53]. The fungi most frequently isolated in respiratory bronchiectasis samples are *C. albicans* (30–45%) and *A. fumigatus* (7–24%) [4,54]. Table 1 shows the yeasts and filamentous fungi most frequently isolated from respiratory samples in bronchiectasis. Although the role of *Candida* spp. in bronchiectasis has not been well studied, Máiz et al. (2015) found that long-term antibiotic therapy was associated with isolation of this fungus [55]. The risk factors for positive cultures of *Aspergillus* spp. were older age, greater lung disease severity, long-term antibiotic therapy, and greater sputum purulence [24,53,54,55,56,57]. *A. fumigatus* isolates have been reported in various clinical situations with highly differing severity in bronchiectasis, from ABPA (especially in patients with asthma as an immune hyper-reactivity) to an invasive form, particularly in bronchiectasis associated with severe immunodeficiency [56,57,58].

### 2.4. Asthma

Asthma is an inflammatory disease with few studies about the microbiome. The composition of the airway fungal microbiota (mycobiome) and its relationship to clinical features are, therefore, unclear [59]. The prevalence of fungal isolates in the airways of patients with asthma is unknown, and most studies on the subject have been conducted to assess the mechanisms and hypersensitivity to fungi such as *Alternaria*, *Cladosporium*, and *Aspergillus*. The involvement of the mycobiome in the development of asthma, in symptom severity, and in their persistence is also unknown [60,61]. In relation to the hypersensitivity mechanisms, 0.7–3.5% of patients with asthma develop ABPA, a disease associated with hypersensitivity to *Aspergillus*, which has a complex pathogenesis in which various immunological mechanisms are involved, such as immediate hypersensitivity (type I), antigen–antibody complexes (type III), and the response mechanisms of inflammatory cells such as eosinophils (type IV-b) [11,62]. A relationship has also been found between exposure to fungi at home and the presence of exacerbations [60]. Sharpe et al. (2015) identified an important association between the presence of *Cladosporium*, *Alternaria*, *Aspergillus*, and *Penicillium* in samples collected at home with the exacerbations in adult patients with asthma, which suggests that environmental fungal exposure leads to asthma exacerbations and has implications for disease severity and management [63].

## 3. Studying the Human Mycobiome Using Next-Generation Sequencing

### 3.1. Introduction

In the last two decades, massive sequencing, also known as NGS or second-generation sequencing, has become popular in the field of microbiological research and diagnostics. This technology makes it possible to identify and understand the distribution of microorganisms in complex ecosystems without the need for culture (Figure 1 shows the algorithm for the study of the mycobiome of respiratory samples). The technology has been applied in studies related to human health, and although it has become standardised for studying bacteria, there is less experience with fungi. The term mycobiome was first used in 2010 [64], and the first studies were conducted shortly thereafter [65,66]. Thanks to genetic fingerprinting, we now know that the diversity of the human mycobiome is greater than was expected because many species have not yet been cultivated [67,68]. Fungi are not generally included in studies characterising the human microbiota, given the fungi’s small representation (0.1% of the microorganisms inhabiting the body) and their supposedly limited pathogenic role, although important studies have been published in recent years in relation to the digestive and respiratory tracts, including in patients with CF [54,69,70,71,72]. The study of the interactions between fungi and bacteria has also been gaining prominence [73], as has the study of the global microbiota metabolism and its impact on health. The modulation of the ecosystem composition and, above all, the particularities of the terms eubiosis and dysbiosis remain unresolved issues [74,75].

### 3.2. Next-Generation Sequencing Strategies

The first second-generation sequencing platform was the 454 (Roche, Basel, Switzerland), which is based on pyrosequencing but has been obsolete since 2013. The next to appear was the Ion Torrent PGM (Thermo Fisher Scientific, Waltham MA, USA), which is based on semiconductors and is now largely displaced by the platforms developed by Illumina (e.g., MiSeq, NextSeq, and NovaSeq), which use fluorescent reversible terminators. Illumina technology produces a large number of high-quality reads but with limited size (300 bases at most). However, with a paired-end approach that includes both read directions, 550 bp can be obtained with acceptable quality and throughput. Most of the bioinformatics software developed for the analysis of data obtained by NGS has been designed for this technology, given that it has been the dominant technology in the last 5 years at a global level.

Third-generation sequencing has been around for several years and aims to obtain long sequences by single-molecule sequencing, sacrificing high throughput (and, thus, increasing the error rate). The PacBio and Sequel (Pacific Biosciences, Menlo Park, CA, USA) platforms use a zero-mode waveguide, whereas the Oxford Nanopore MinION, GridION, and PrometION (Oxford Nanopore Technologies, Oxford, UK) use nanopore-based technology. The latter are simple, portable, and inexpensive devices that can reach read lengths of up to 2.4 Mb. The MinION device connects via USB to a laptop and allows sequencing to be performed anywhere, including at the patient’s bedside but with a moderate error rate (6–12% of all nucleotides) [76].

DNA sequencing can serve several purposes: (1) to reveal the species-level identity of an isolate; (2) to determine the taxonomic composition of an entire ecosystem; (3) to assess the genetic diversity of isolates of the same species through highly conserved genes; (4) to decipher the entire genome of a particular isolate (whole genome sequencing [WGS]); and (5) to estimate minority populations with point mutations. Variations also include the use of (1) total DNA from a sample (shotgun strategy); (2) prior amplification of a gene (phylogenetic markers are used, in general, to decipher its taxonomy: 16S rDNA for bacteria and ITS for fungi); and (3) DNA from pure isolates (WGS and minority populations).

WGS studies make it possible to define clonal transmission by analysing single nucleotide polymorphisms and core genome multilocus sequence typing. They are also used to differentiate the core genome of the species from the accessory genome, helping to identify transmission routes, especially in nosocomial infections, and to identify the virulence or antifungal resistance determinants of each isolate.

### 3.3. Amplification Targets

If we focus on the study of microbial communities, the factors that determine the best sequencing target are taxonomic resolution, coverage, accuracy, and amplicon length. Although an internal 16S rDNA gene fragment is always used for bacteria, there is no single strategy for fungi. Early studies have focused on the 18S small subunit or the 28S large subunit rDNA, whereas recent studies prefer the internal transcribed spacer (ITS) for its higher taxonomic discriminatory power [77]. The intergenic spacer ranges in length from 500 to 700 bp and is differentiated into ITS1 and ITS2 subregions, which, in turn, are separated by the conserved 5.8S region [78]. A number of authors have proposed the use of degenerate primers (gITS7/ITS4 and 5.8S-Fun/ITS4-Fun) to improve coverage and specificity, but most studies on the subject use the non-degenerate primers ITS1F/ITS2 (targeting the ITS1 region). Primer selection should be carefully performed because certain primers, such as ITS1 and ITS1-F, are biased towards the amplification of basidiomycetes, whereas others, such as primers ITS2 and ITS4, are biased towards overrepresentation of ascomycetes [79,80,81].

### 3.4. Bioinformatics Analysis

The strategy of the bioinformatics analysis of the sequences has an important impact on the results. The nature of each study, with its added biological factors as well as the various protocols for processing samples and sequencing platforms on the market, prevent the standardisation of a generic “pipeline”. Communication between the bioinformatician and clinician/researcher is, therefore, essential in this case. Broadly speaking, bioinformatics analysis involves three stages.
**Primary analysis**, which includes the transformation of the sequencer readings into base calls with their associated quality data. This is a closed process that is automatically performed by the sequencer itself and typically includes the demultiplexing process, whereby each read is assigned to its source sample (thanks to the unique barcode that is added to each library before sequencing). The outputs of this process in the case of Illumina platforms are FASTQ files.**Secondary analysis**, which refers to the curation of sequences and their counting and classification and typically includes the following:
The trimming of adaptor and primer regions as well as conserved regions flanking the target region (ITS). The latter is a special feature that only applies when sequencing ITS. The size of this region varies according to the fungal species, requiring the use of tools that detect these ends for each particular sequence and cut at that point.Filtering out low-quality sequences (a factor given by the sequencing platform as a percentage of safety for each nucleotide).Filtering of sequences of lower-than-expected size.Merging of forward and reverse sequences (in the case of Illumina, paired-end protocols).The search and generation of representative sequences (amplicon sequence variants (ASVs)) and count tables per sample.The taxonomic assignment of ASVs. The main reference databases for the taxonomic assignment of fungi are UNITE, INSDC, SILVA, Warcup, and FindFungi.The elimination of ASVs not assigned to the kingdom Fungi.Data normalisation: Process that normalises the number of reads per sample for later comparison.**Tertiary analysis**, which begins once the processing of raw reads to achieve the ASVs is completed. This stage includes the statistical analysis of the sequencing data considering the clinical variables of the study samples. First, the alpha diversity (richness of each sample) is analysed, generally using the Shannon, Chao1, and Faith-PD indices. These parameters consider the number of species, their distribution, and, in the case of the latter, their phylogenetic relationships. The alignment of the ITS region is not useful for inferring the evolutionary distances between very distant species, although there are methods to partially overcome this limitation (https://github.com/JTFouquier/q2-ghost-tree, accessed on 10 April 2022). Moreover, beta-diversity analyses compare the composition between groups of samples (established according to the clinical variables collected) using indices that represent their similarity/dissimilarity, such as Bray Curtis, Jaccard, and UniFrac. These differences can be subjected to statistical analysis to assess their significance and/or represented graphically (e.g., via principal coordinate analysis). Differential abundance analyses can also be performed to identify which species primarily explain the differences between groups, using the linear discriminant analysis effect size tool based on linear discriminant analysis, as an example. Various platforms (either developed in a Linux environment or as web tools) facilitate the analysis, integrating the different tools and, above all, their statistical significance. These platforms include QIIME2 [82], mothur [83], CloVR-ITS [84], CONSTAX [85], and HumanMycobiomeScan [86].

### 3.5. Limitations

The massive sequencing strategy has provided insight into the true complexity and composition of the human microbiota in general and the mycobiome in particular; however, it is also important to know and be aware of the limitations of these techniques.

Sample collection and processing is always a key point to obtain meaningful results in sequencing studies. Collection should be performed, whenever possible, under sterile conditions, avoiding contamination by environmental fungi and by the personnel collecting the sample. The best option is to process the sample as soon as possible, but given that this is not usually feasible, it is recommended that samples be frozen immediately (at −80 °C).

DNA extraction can also affect the final results. The fungal wall is often difficult to lyse, limiting DNA release and its subsequent amplification and sequencing. The use of physical means to break the fungal wall, such as glass beads, is effective but can lead to DNA fragmentation. The presence of glucans, chitin, mannans, and glycoproteins requires additional steps for wall dissolution/fragmentation. Manual extraction with phenol/chloroform appears to obtain the best results. While the extraction methods affect DNA yield and quality, their impact on mycobiome composition and diversity seems minor; however, it is always important to consider the methodology of DNA extraction when comparing studies [74,79,87,88].

During the polymerase chain reaction process, all genes present in the sample are amplified, regardless of whether the fungus is metabolically active or dead. DNA from external contamination can also be amplified. To minimise this limitation, sample pretreatment with propidium monoazide is recommended to avoid the over-representation of non-viable fungi or free DNA [89]. It is also essential to include appropriate negative controls throughout the DNA extraction, amplification, and sequencing processes to rule out external contamination. Many studies have proposed the inclusion of an artificial fungal community (mock community) in each experiment to corroborate the quality and reproducibility of sequencing and bioinformatics.

With these processes, the identity and distribution of fungi can be determined; however, their pathogenic or commensal nature cannot be determined, nor can an absolute quantification be performed due to the limitation of the amplification process in which the most frequent fungi is always much more amplified and minority populations are underestimated.

Although these processes are increasingly integrated in microbiology laboratories, they require computer clusters, high-capacity computers, and specialised bioinformatics for their analysis, which is not within the reach of all laboratories. Undoubtedly, the largest problem is the absence of criteria to define a “normal” mycobiota, and there is no definition for when it is altered; we simply analyse the statistical differences between its composition in patients versus healthy controls. This lack of criteria can be partially explained by the methodological heterogeneity of microbiota-based studies, which could be overcome through the use of guidelines for human microbiome research [90].

## 4. Structure and Composition of the Lung Mycobiome

### 4.1. Conventional Culture versus Next-Generation Sequencing

Fungal culture from respiratory samples is a useful conventional method for the isolation, identification, and antifungal susceptibility testing of fungi implicated in the pathology of CRDs. However, this classical approach has certain limitations. First, fungal culture from respiratory samples is not routinely performed in some microbiology laboratories, and there are no standardised protocols [56,91]. It has also been shown that the usual seeding on Sabouraud-chloramphenicol agar and the 5-day incubation times used in most microbiology laboratories might underdiagnose the presence of fungi in the lower airways [56,91,92]. Alternative protocols have been proposed to increase the yield of fungal culture, including homogenisation of samples, seeding of a larger inoculum, longer incubation times, and the use of enrichment and/or selective media for the growth of specific fungi, such as *E. dermatitidis* and *S. apiospermum* [93,94]. Although these modifications might increase the culture yield for certain fungal species, it is likely that there are many other species in the lung microbiome that cannot be detected by conventional culture. In addition to the fact that many patients have negative cultures and that the clinical significance of fungal isolation is still controversial, the introduction of a large number of culture media for diagnosing fungal infections in CRD might not be cost-effective.

NGS has been shown to detect a much larger number of fungal species than conventional culture, many of which are difficult or impossible to cultivate using conventional techniques, such as *Malassezia* species [54,66,73,95,96,97]. Many of the fungi detected, however, could be merely transient colonisers from the continuous inhalation of conidia from the environment, which are detected thanks to the high sensitivity of these techniques [92]. The high sensitivity of NGS and the large number of species detected make it more difficult to elucidate the role of fungi in the pathology of CRDs. Therefore, when studying the mycobiome, it is important to stratify patients according to their clinical stage and/or suspected fungal infection. It is also important to perform longitudinal studies in which changes in mycobiome composition over time are analysed and related to changes in the bacterial communities and the patient’s clinical status. In these cases, changes in the mycobiome have been observed, indicating that fungi might play an important role in the pathophysiology of CRDs [15,95].

### 4.2. Healthy Individuals

The lower respiratory tract of healthy individuals has, for many years, been considered a sterile body compartment. However, NGS techniques have revealed that there is a complex community of bacteria with a similar structure and composition to that found in the upper respiratory tract (URT), albeit in much smaller numbers, the most common phyla being Bacillota (formerly Firmicutes) and Bacteroidota (formerly Bacteroidetes) and the most common genera being *Prevotella*, *Streptococcus*, and *Veillonella* [14]. It is thought that bacteria reach the lung from the URT by microaspiration and that their population is controlled by the body’s mechanical and immunological defence mechanisms. In the case of fungi, the origin can be either the URT (e.g., *Candida* spp.) or the continuous inhalation of spores that are present in the environment (e.g., most filamentous fungi) [48,66,73,92]. Ascomycota and Basidiomycota are the most commonly identified phyla, and *Candida*, *Saccharomyces*, *Penicillium*, *Cladosporium*, and *Fusarium* are the most common genera [73,98,99,100].

There is growing evidence that, as with the gut microbiota, the bacterial communities in the lungs of healthy individuals play a role in immune development and tolerance to microbial antigens via metabolite production by living organisms and pattern recognition receptors by live and dead bacteria [14]. Segal et al. (2016) showed that when oral commensal microbiota were detected in the bronchoalveolar fluid (BALF) of healthy individuals, they exhibited a less robust TH_17_/neutrophilic immune response than individuals in whom they were not detected [101]. The immunomodulatory role of the microbiota has also been demonstrated in animal models. In germ-free neonatal mice, exaggerated lung inflammatory responses to allergens were observed and were subsequently reduced as the lower respiratory tract became colonised with bacteria [102,103]. Moreover, the intratracheal application of oral commensals in mice induced a TH_17_ response that conferred them resistance to *Streptococcus pneumoniae* infection [104]. Similarly, it was shown that antibiotic therapy in mice makes them more susceptible to respiratory infections after pathogen exposure [105]. Taking these data as a whole, it appears that lung microbiota play an important immunomodulatory role in the lower airways, despite their low numbers. However, whether the lung mycobiome contributes in the same way to normal physiology remains unknown [14].

### 4.3. Patients with Chronic Obstructive Pulmonary Disease

There have been few studies using NGS to analyse the pulmonary mycobiome of patients with COPD [106,107,108,109]. Recently, Martinsen et al. (2021) performed a large study on the oral and lung mycobiomes (BALF samples) of 93 patients with COPD to compare their structure and composition with those of healthy individuals (*n* = 100) and to evaluate the effect of inhaled steroids on these variables [109]. As in other CRDs, the lung mycobiome in patients with COPD is dominated by *Candida* species (*C. albicans* being the most prevalent), followed at a great distance by a variety of yeasts and filamentous fungi, with *Malassezia*, *Sarocladium*, *Penicillium*, *Aspergillus*, and *Fusarium* being the most important genera [109]. These authors found no substantial differences in mycobiome composition or diversity between cases and controls, nor did they find a clear effect of inhaled steroids on the mycobiome structure and composition in patients with COPD. However, this study had limitations, such as not being a longitudinal study and not analysing the possible interactions between fungi with bacteria and with the host immune response. Moreover, the patients included in the study were in a stable phase of disease, with those having signs of respiratory infection/exacerbation being excluded [109]. Su et al. (2015) found differences in mycobiome composition during acute exacerbations in a longitudinal study of six patients with COPD; however, these were specific to each patient, and, therefore, a clear trend could not be found [107]. Interestingly, other studies have found that specific fungal species could have a relevant impact on COPD. A recent study identified two distinct COPD subtypes according to their fungal colonisation/infection [108]. One subtype is associated with increased symptoms and *Saccharomyces* dominance, whereas the other subtype was associated with very frequent exacerbations and higher mortality, characterised by *Aspergillus, Penicillium*, and *Curvularia* species, with a concomitant increase in serum-specific immunoglobulin (Ig) E levels against the same fungi [108]. Similarly, *Pneumocystis jirovecii* has been found to be over-represented in HIV-positive patients, especially in those with COPD, suggesting a possible role of this unculturable fungus in the pathogenesis of the disease in this particular population [106].

### 4.4. Patients with Non-Cystic Fibrosis Bronchiectasis

The pulmonary microbiome in bronchiectasis is altered compared to healthy individuals [52]. It has been well-established that the presence of microbiome-predominant *P. aeruginosa* is associated with higher bronchial and systemic inflammation, greater clinical and functional lung severity, and poor outcomes in bronchiectasis [110]. However, very little is known about the mycobiome in bronchiectasis, although the combination of *A. fumigatus* and *P. aeruginosa* has been linked with a more intense immune response [75]. Recently, Cuthbertson et al. (2021) studied the mycobiome of 42 patients with non-CF bronchiectasis and 134 patients with CF, stratifying them according to clinical and analytical criteria into patients with ABPA, chronic necrotizing pulmonary aspergillosis, and fungal bronchitis and patients with no evidence of fungal infection [54]. In patients with non-CF bronchiectasis, the authors described a more diverse mycobiome (although less abundant than in patients with CF), dominated by species of the genera *Candida* (mainly *C. albicans*), *Aspergillus*, and *Penicillium*. In contrast to patients with CF, patients with non-CF bronchiectasis and ABPA had a mycobiome dominated by *A. fumigatus*. This dominance was also observed in patients with chronic necrotizing pulmonary aspergillosis, with which this species was significantly associated [54]. Globally, *A. fumigatus*, *E. dermatitidis*, and *S. apiospermum* were detected in 96.9%, 28.1%, and 21.9% of patients with non-CF bronchiectasis by NGS, whereas cultures did not detect any filamentous fungi. This study demonstrates the limited usefulness of cultures in the diagnosis of fungal infection in bronchiectasis [54].

### 4.5. Patients with Cystic Fibrosis

The first characterisation of the mycobiome (together with the microbiome) in patients with CF was performed by Delhaes et al. (2012) [66]. This and subsequent studies discovered fungal communities dominated by the genus *Candida*, with *C. albicans*, *C. parapsilosis*, and *C. dubliniensis* being the most abundant species [54,66,73,92,95]. The origin of these yeast-like fungi is probably the URT, and they probably colonise the lower airway by microaspiration events [48,95]. Interestingly, *Malassezia* species are frequently detected in respiratory samples from patients with CF [66,73,89,95]. This basidiomycetous yeast is not detected by routine culture of respiratory samples due to its lipophilic nature. Interestingly, the yeast has recently been associated with pulmonary exacerbations in patients with CF as well as with other chronic inflammatory conditions, such as asthma [73,96]. The origin of filamentous fungi is likely the inhalation of airborne conidia, and studies have found a great diversity of species dominated by the genera *Penicillium*, *Aspergillus*, *Fusarium*, *Cladosporium*, and *Eurotium*, among others [54,66,73,89,95]. It is necessary to discern, however, whether they constitute stable lung communities or whether, on the contrary, they are transient colonisers from the occasional inhalation of conidia and are detected thanks to the high sensitivity of NGS. Kramer et al. (2015) conducted a sequential study of the mycobiome of patients with CF over a 2-year follow-up during which the authors reported a continuous increase in the number of newly detected taxa of filamentous fungi as more samples were analysed [92]. This increase was not observed for bacterial taxa, in which a plateau in newly described species was reached. When analysing air samples from the patients’ environment, the authors found a similar composition of filamentous fungi, concluding that these organisms correspond mostly to transient colonisers [73,92]. *A. fumigatus*, *S. apiospermum*, and *E. dermatitidis* could be an exception to this rule, given that they are detected repeatedly in patients with CF in this and other sequential studies [73,92]. The pathogenic role of these three species has also been demonstrated by Cuthbertson et al. (2021), who described a less diverse mycobiome in patients with CF with fungal bronchitis than in patients with no evidence of fungal infection, which was dominated by these three pathogens [54]. The authors found no association, however, between *Aspergillus* detection and ABPA, suggesting a possible role of other fungi, such as *Candida* in fungal sensitisation [54].

Pulmonary exacerbations (PEs) are periods in which lung function is significantly reduced and intravenous antibiotic therapy is required, despite which baseline lung function is not restored in up to 25% of patients [111]. The onset of PEs is not well understood but appears to be due to the presence of a transient virulent bacterial community (attack community) dominated by anaerobic bacteria with fermentative metabolism. In the stable phase, however, a stable bacterial community (climax community) is found that is predictive of long-term patient outcomes [73,89,112]. Fungal communities have recently been included in this climax/attack model thanks to new advances in statistical network inference tools that permit the analysis of microbial communities as a whole during PEs (Figure 2). Soret et al. (2020) found an association between anaerobic attack communities and *Aspergillus*, *Candida*, and *Malassezia*, probably because they create advantageous conditions for the growth of these fungi. Moreover, the authors found a significant association between *Aspergillus* and *Malassezia* and PEs. In contrast, *Scedosporium* spp. were found to be part of the climax communities and were significantly associated with poorer lung function (Figure 2) [73]. Further studies of this type with larger numbers of patients in various clinical situations are likely to provide more data on the true pathogenic potential of fungi in CF.

### 4.6. Patients with Asthma

Recent studies have related asthma severity to the pulmonary mycobiome, with differences in fungal isolates in patients with severe asthma, ABPA, asthma with fungal sensitisation, and mild asthma, highlighting that patients with severe asthma have a relative abundance of *Aspergillus* in the airways 15 times greater than those with mild asthma [113]. A number of studies have also indicated the possibility of a complex interaction between the pulmonary mycobiome and the immune system, such that certain fungi could play a role as adjuvant factors that can increase the Th2 allergic response [96,114]. A number of authors have also reported an increased incidence of fungi in the airways of patients with obesity and asthma, an important finding given that obesity acts as a trigger and worsening factor for asthma [115].

Other studies have compared the airway’s mycobiome in patients with asthma compared with healthy controls. Van Woerden et al. (2013) found that in patients with asthma, the sequences of *Psathyrella candolleana*, *Malassezia pachydermatis*, *Termitomyces clypeatus*, and *Grifola sordulenta* were more prevalent, drawing attention to the presence of the fungus *Malassezia pachydermatis* in the airway of patients with asthma, given that this pathogen is known to be associated with atopic conditions, including atopic dermatitis [96,97]. Sharma et al. (2019) conducted a study linking the mycobiome with various asthma endotypes (high Th2 and low Th2 responses) defined by the eosinophil number. In BALF samples, the authors found an increased presence of *Fusarium*, *Cladosporium*, and *Aspergillus* in patients with a high Th2 response and of *Cladosporium* and *Fusarium* in patients with asthma without atopy. The authors also observed a relationship with clinical variables, such FEV_1_ values, which were related to sequences of *Alternaria*, *Aspergillus*, and *Penicillium*, the use of inhaled corticosteroids with *Alternaria* and *Cladosporium*, and treatment with oral corticosteroids with *Cladosporium* [99].

## 5. Role of the Mycobiome in Chronic Pulmonary Diseases

### 5.1. Direct Implications

The real role of the mycobiome in chronic lung infection has not been well-defined except for a few filamentous fungi, such as *Aspergillus* spp. and *Scedosporium* spp., and yeasts, such as *C. albicans*. In many cases, the leading role in lung function impairment is attributed to bacterial pathogens that were classically sought after in microbiological cultures and referred to in a number of publications, at least in chronic bronchitis, as “potentially pathogenic microorganisms”. Fungi are also downplayed because of uncertainty as to whether they are considered mere colonisers or even contaminants. The application of high-throughput sequencing techniques has provided new data on the presence of fungal communities in the respiratory tracts of patients with chronic lung infections but has also shown difficulties in interpreting the results [116]. One of the largest studies that was conducted with 403 patients with chronic bronchitis and sequential sampling (in the stable phase, during exacerbation, and 2 weeks later) demonstrated a relative stability of the mycobiome composition and diversity in these patients, despite treatment with oral antibiotics or corticosteroids during exacerbations [108]. This study partly contradicts previous studies that showed, as with the bacterial microbiome, a decrease in diversity during exacerbations [66,117].

The isolation of fungi in the stable phases of patients with respiratory disease might indicate an incipient stage, the possible future progression of the disease and a transition to a clinical worsening and deterioration of lung function, or simply its presence as a coloniser. *Aspergillus* has been associated with a high proportion of progression to invasive aspergillosis (almost 22%) and increased mortality risk during disease progression [118]. Risk factors for patients with COPD include exacerbations in the previous year, the concomitant isolation of *P. aeruginosa*, and the use of inhaled corticosteroids [118,119].

### 5.2. Host and Bacterial Interactions with the Mycobiome

The fungal–bacterial interaction and its relationship with the host has been studied in various articles and has been extensively reviewed in relation to the different niches in which it can coexist [113,120,121]. This interaction could favour the colonisation by fungi, bacteria, or both, either sequentially or simultaneously as a synergism. Niche exclusion (or antagonism) can also occur, whereby the presence of fungi prevents colonisation by bacteria and vice-versa or determines displacement when colonisation by a different fungi or bacteria occurs. A final case would be the additive model in which colonisation occurs only when both microorganisms are present [120,122]. Except for cases in which there is niche exclusion (especially in co-colonisation processes), virulence can be enhanced by fungi–bacteria interactions or with the host, either directly, by metabolic products, or by environmental changes (e.g., pH and oxygen tension). In addition, biofilm formation hampers interaction with the immune system and, where applicable, antimicrobials (Figure 3). One of the most studied models of this interaction is CF and, in particular, *P. aeruginosa* and *Aspergillus* spp. It has been known for years from in vitro studies that the production of certain metabolites such as pyocyanin and phenazine by *P. aeruginosa* inhibits the growth of *A. fumigatus* and *C. albicans* [123]. Reciprocally, *Aspergillus* also produces an antagonistic effect on *P. aeruginosa*, even when it develops into biofilms [124]. Moreover, farnesol production by *C. albicans* affects *P. aeruginosa* development and quorum-sensing signals [120]. These in vitro observations do not always present themselves in the same way in experimental animal trials; thus, extrapolation of their clinical significance is uncertain. Interactions would also not occur in the same way with other Gram-negative microorganisms and between these and different fungi. There is, therefore, no general model that allows for universal conclusions to be drawn.

The interaction between fungi and the host can occur directly or through the metabolites produced during the colonisation and infection phases. In the latter case, the interaction can even be at a distance [125]. Animal models of asthma have shown that the colonisation of the intestinal tract by *C. albicans* also leads to lung inflammation [126], and prostaglandin E2 or interleukin-1 could be responsible for this process [125,127].

At the respiratory mucosa level, *Aspergillus* spp. is a potent inducer of interleukin-22, which in turn determines the induction of defensins, peptides with antimicrobial activity that influence the composition of the pulmonary bacterial microbiome, including *P. aeruginosa*. *Aspergillus* spp. also elicits a Th2 and Th17 response and an increase in macrophages in the lung. In the case of *C. albicans*, the response is due to Th17, which is intimately related to the expression of prostaglandin E2 induced by its metabolites [127].

The interaction of *Aspergillus* spp. can generate ABPA, which is closely related to asthma and is characterised by local inflammation, increased Th2 cytokine response, IgE, and eosinophilia [128]. ABPA is caused by a hypersensitivity reaction with *Aspergillus* colonisation of the bronchial tree, which is also influenced by lipoxygenase expression by *Aspergillus*, which shows homology to human 5-lipoxygenase, a protein with enzymatic activity that is involved in asthma [125].

## 6. Conclusions and Future Perspectives

Studies performed to date point to the presence of a lung mycobiome in the lower airways of healthy individuals and patients with CRDs. Among the diseases involved, the mycobiomes of asthma and CF patients have been the most studied. Generally, the lung mycobiome is dominated by *Candida* yeasts and a highly variable fraction of filamentous fungi, many of which might be transient species that are inhaled from the environmental air. Although this is an emerging field of study, high-throughput sequencing has much more sensitivity than conventional cultures for detecting fungal infection/colonisation of the lower airways, which, together with new statistical and bioinformatics analyses, have linked the detection of fungi, such as *Malassezia* and *Aspergillus*, with the pathology of CRDs. More studies are needed to elucidate the role of fungi in the pathophysiology and prognosis of CRDs, preferably with a follow-up longitudinal design in which mycobiome and bacteriome composition would be compared with changes in patients’ clinical condition over time.

## Figures and Tables

**Figure 1 microorganisms-10-00810-f001:**
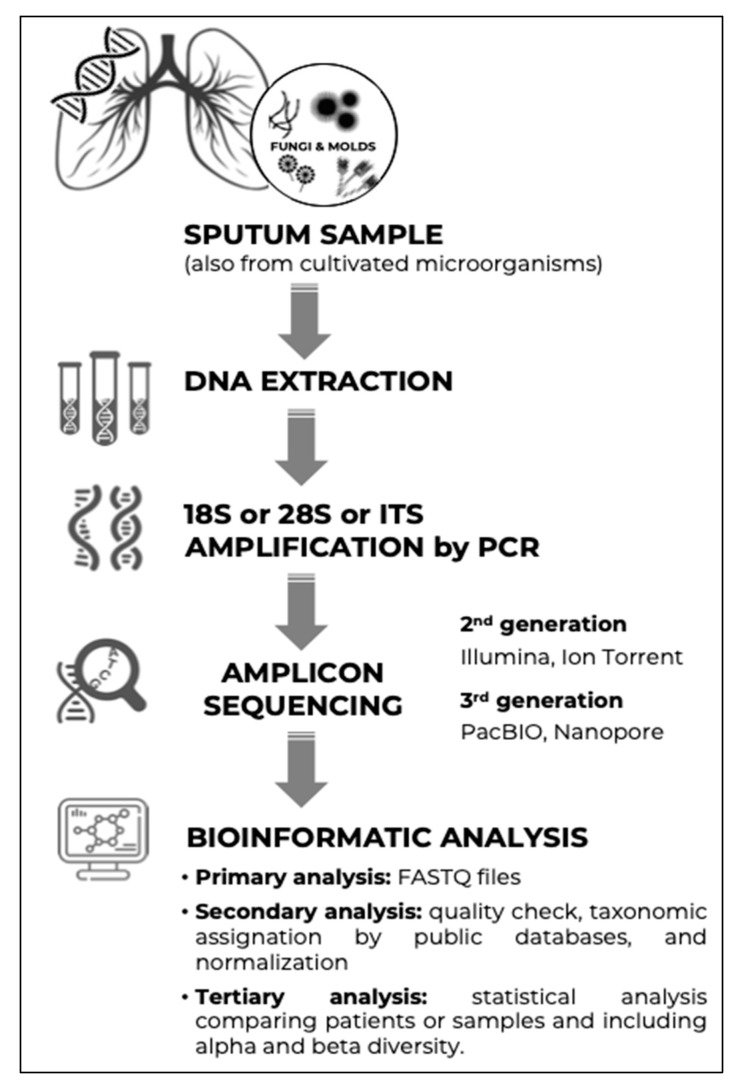
Steps for the study of the mycobiome in respiratory samples.

**Figure 2 microorganisms-10-00810-f002:**
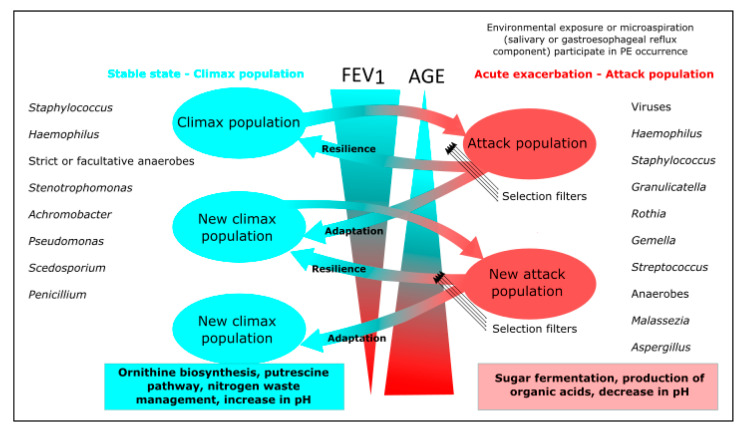
Adaptation of climax/attack model (CAM) in cystic fibrosis. According to the CAM, there are two dynamically evolving bacterial populations in the CF lung, both being potentially composed of anaerobes. Environmental exposure or microaspiration events change microbiome structure, producing an attack population that triggers pulmonary exacerbations (PEs). The microbial community can return to its original state (resilience) or move to a new stable (climax) community with a different microbiome composition (adaptation). Whether resilience or adaptation occurs depends on the disruptive forces of the attack population and its ability to pass through selection filters. These include changes in nutrient sources, oxygen pressure, pH, microorganism growth and virulence, host immune response, and antimicrobial treatment. These filters participate in the selection of the best-suited population to new airway remodelling, in a circular relationship. According to carbon source, the climax population uses amino acids and produces ammonia, whereas the attack population ferments sugar and produces acids. This could explain the association between *Malassezia* and anaerobes, given that this yeast is unable to ferment sugars, could take advantage of these organic acids as a carbon source (cross feeding), and is also able to grow at low pH. On the other hand, the association between lower FEV_1_ values and *Scedosporium* could be explained by its belonging to an advanced disease climax population, thanks to its ability to use a wide range of nutrients (including ammonia fermentation) and to its high resistance to antifungals. Modified from Soret P, et al. Sci Rep. 2020 [73]. CC BY 4.0.

**Figure 3 microorganisms-10-00810-f003:**
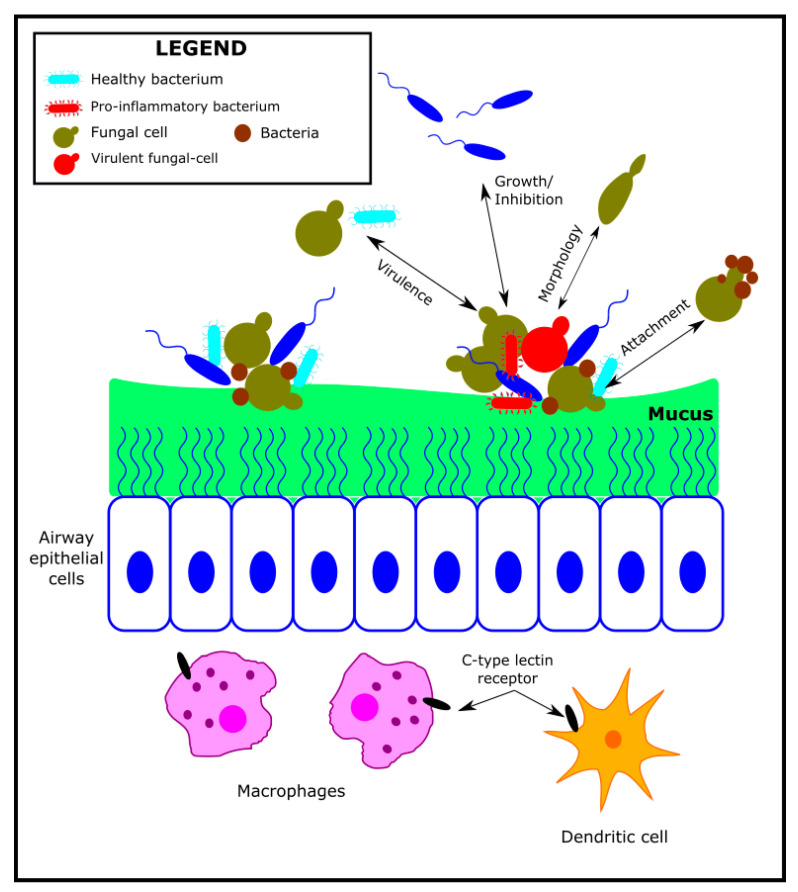
Interaction between bacterial microbiome, mycobiome, and host immune system in the airways. Bacteria and fungi coexist in the lower airways as polymicrobial biofilms attached to the mucosa. Fungi can selectively induce or inhibit the growth of various bacterial taxa, increase the expression of bacterial virulence factors, alter bacterial morphology, and act as attachment sites for bacteria. Similarly, bacteria can also alter fungal growth, virulence, morphology, and attachment. In addition, C-type lectin receptors on macrophages and dendritic cells, such as dectin-1 and Mincle, can sense fungi and mediate host inflammatory responses. Modified from Zhang et al. (2017) [113]. CC BY 4.0.

**Table 1 microorganisms-10-00810-t001:** Fungi most frequently isolated from respiratory samples from patients with non-cystic fibrosis bronchiectasis.

Yeasts	Filamentous Fungi
*Candida albicans*	*Aspergillus fumigatus*
*Candida glabrata*	*Aspergillus niger*
*Candida parapsilosis*	*Aspergillus terreus*
*Saccharomyces cerevisiae*	*Scedosporium apiospermum*
*Trichosporon beigelii*	*Penicillium* spp.
*Exophiala dermatitidis*	*Fusarium* spp.

## Data Availability

Not applicable.

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
