# Peer review of "The Human Mycobiome in Chronic Respiratory Diseases: Current Situation and Future Perspectives"

_microorganisms, 2022, doi:10.3390/microorganisms10040810_

Round 1
Reviewer 1 Report
The manuscript covers an interesting topic. It is well written, however no methodological section is provided. Please, add information on how the review has been conducted.
Please, specify in the title that it is a review, also specify what type of review
Please, add a figure regarding the section 3 "Studying the human mycobiome by Next-Generation Sequencing"
Also, plwasw considwe to add a graphical abstract. The content of the manuscript is interesting but not easy to follow. Adding tables and figures may improve the readability.
Author Response
Dear reviewer
The authors would like to thank you for your review work and your positive comments which have helped to improved the article. Now, I will try to answer to your comments point-by-point:
The manuscript covers an interesting topic. It is well written, however no methodological section is provided. Please, add information on how the review has been conducted.
We respectfully feel that a methodological section is not necessary in this type of work, as it is a literature review and not a systematic review. This work has been made by experts in different clinical and laboratory fields, who have included the literature that they considered to be most relevant. After reviewing similar papers submitted to this journal, we have not found a methodological section in any of them. We, therefore, have decided not to add this section in this work.
Please, specify in the title that it is a review, also specify what type of review
Again, respectfully, we think it is not necessary to specify this in the title, as it is already specified in the journal's template. In our opinion, it would lengthen the title without providing any relevant information.
Please, add a figure regarding the section 3 "Studying the human mycobiome by Next-Generation Sequencing"
We have added a new figure (Figure 1), in which we have summarised the workflow for the analysis of respiratory samples by NGS.
Also, please consider to add a graphical abstract. The content of the manuscript is interesting but not easy to follow. Adding tables and figures may improve the readability.
We think the graphical abstract is a good idea and we have made one to include in the pubblication.
Reviewer 2 Report
This work is well conducted, it does significantly summarize the research around human mycobiome in chronic respiratory diseases.
I recommend the acceptance of the manuscript after language corrections.
Author Response
Dear reviewer
The authors would like to thank you for your review work and your comments, which have served to improve the article. We have sent the paper to a scientific English reviewer and have made the corrections suggested by him.
Round 2
Reviewer 1 Report
thank you for the efforts in improving the manuscript